# Temporal and Spatial Water Quality Assessment of the Geumho River, Korea, Using Multivariate Statistics and Water Quality Indices

Changdae Jo *, Heongak Kwon and Seongmin Kim

Nakdong River Environment Research Center, National Institute of Environmental Research, Dalseong-gun, Daegu 43008, Korea; hun7082@korea.kr (H.K.); frogksm@korea.kr (S.K.)
* Correspondence: ship44@korea.kr; Tel.: +82-53-602-2718

**Abstract:** The Geumho River in South Korea passes through a metropolitan area with a high population density and multiple industrial complexes; therefore, the water quality of this river is of significance for human health and economic activities. This study assesses the water quality of the Geumho River to inform river water quality management and improve pollution control using multivariate statistics and the Korean Water Quality Index (KWQI). Principal component and factor analyses identified factors related to organic pollutants and metabolism (principal factor 1) and phosphorus and fecal coliform content (principal factor 2). Based on the results of the cluster analysis, it was classified into four groups in time and three groups in space. Six temporal variables and seven spatial variables were extracted from discriminant analysis results; the most important water quality variables were high during the spring and summer seasons and in the midstream and downstream regions. Temporally, the KWQI was the highest in winter and the lowest in spring; spatially, the KWQI was the highest in the upstream and the lowest in the midstream sections. These results indicate that to improve effectiveness, water management interventions in the Geumho River should focus on the urban midstream section and spring season.

**Keywords:** Geumho River; water quality; multivariate statistics; river water quality management





## 1. Introduction

The water quality of a river is affected by pollution (e.g., from nonpoint sources and non-biodegradable organic matter) generated from natural processes, urbanization, and regional development, often in association with agricultural and industrial activities [1–4]. When the self-purification capacity of a river is exceeded due to the inflow of such pollution sources, environmental problems occur, including poor water quality [5,6]. In the past, organic matter management through biochemical oxygen demand (BOD) regulation was possible because most pollutants were predominantly biodegradable. However, with advances in industry and rapid urbanization, the use and release of inorganic chemicals and non-biodegradable organic matter have increased [7,8]. As a result, the inflow of excessive nutrients to rivers can cause serious environmental problems, including algal blooms, eutrophication, and large-scale fish kills. To effectively manage surface water quality, it is important to control the pollution sources affecting rivers by collecting reliable data, assessing temporal and spatial water quality dynamics, and analyzing the causes of water pollution [9–12].

Various statistical analysis techniques have been used to monitor water quality in rivers, including multivariate statistical techniques (MSTs), such as principal component analysis (PCA), factor analysis (FA), cluster analysis (CA), discriminant analysis (DA), and analysis of variance (ANOVA) [13–17]. These methods inform water resource management by enabling the assessment of key factors affecting water quality and their temporal and spatial dynamics. Indeed, worldwide, many studies have sought to assess river water

quality patterns and dynamics using various statistical techniques and propose measures to rapidly solve pollution and associated water-quality problems [13,18–21].

The water quality index (WQI) is one of the approaches to evaluate changes and trends in the water environment by integrating several parameters into one index [22–24]. By determining and weighting parameters important for water quality assessment, comprehensive water quality status and ratings are presented. WQI plays an important role in evaluating the water quality of rivers, lakes, and dams [3,25,26].

With increasing industrialization and urbanization worldwide, the management of rivers in metropolitan areas is becoming more important [27–29]. In the case of the Geumho River in South Korea, the upstream section is relatively clean, and nonpoint pollution sources discharge into the river under certain rainfall conditions. In contrast, there are multiple pollution sources along the midstream section, which flows through a metropolitan city. However, current water quality management in the Geumho River Basin focuses on BOD and total phosphorous (TP) concentrations only in the downstream section. Therefore, in this study, changes in temporal and spatial water quality characteristics were compared through the statistical analysis of monitoring data from 2017 to 2020.

In this study, the water quality characteristics of the Geumho River were assessed using various MSTs, along with the Korean WQI (KWQI). The combination of these techniques could effectively evaluate the water quality characteristics relative to their individual use. The key purpose of this study was to (1) identify major factors affecting the river water quality using PCA/FA, (2) perform spatial and temporal classification using CA, (3) identify the characteristics of river water quality by extracting major spatial and temporal parameters using DA, and (4) evaluate the condition of river water quality using KWQI. The data analysis process used for this study is illustrated in Figure 1. The study provides a useful tool for identifying the sources and dynamics of pollutants found in rivers passing through various metropolitan areas, aiding the development of appropriate and targeted water quality control measures.

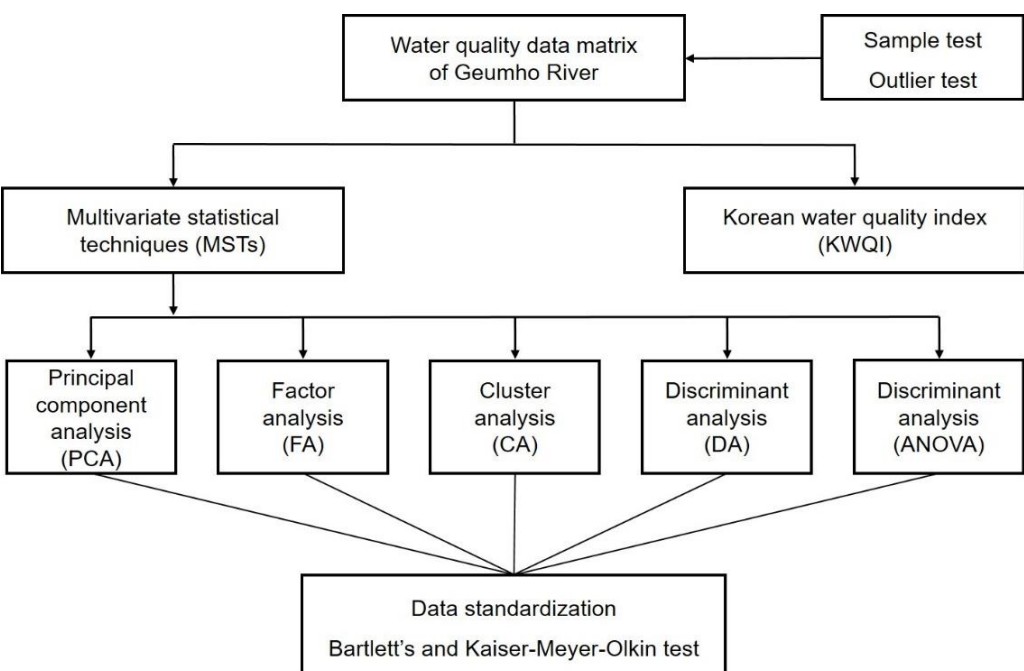

**Figure 1.** Flow chart of data analysis on the Geumho River.

## 2. Materials and Methods

### 2.1. Study Watershed

The Geumho River originates in Pohang City and passes through Daegu Metropolitan City via Yeongcheon Dam, Yeongcheon City, and Gyeongsan City (Figure 2).

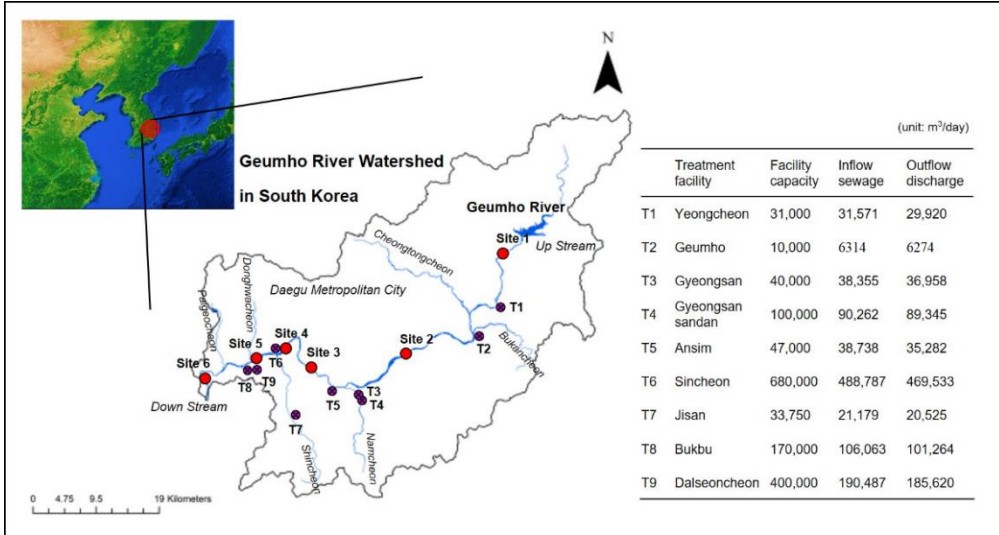

**Figure 2.** Locations of water monitoring sites and sewage and wastewater treatment facilities in the target watershed [30]. Red point, monitoring sites (S1 to S6); purple point, sewage and wastewater treatment facilities (T1 to T9).

There are more than 20 main inflow tributaries to the Geumho River, including the Shinryeongcheon, Bukancheon, and Cheongtongcheon in the upstream area and the Omokcheon, Namcheon, Shincheon, Palgeocheon, Dalseocheon, and Ieoncheon in the downstream area. These combine and flow into the Nakdong River. The average annual precipitation in the Geumho River watershed is 1089 mm, which is significantly lower than the national average of 1159 mm (https://data.kma.go.kr: (accessed on: 1 May 2021)). In particular, water quality deteriorates during the relatively dry season (January to April) due to a lack of instream flow, whereas rainfall is concentrated during the flood season (July to September) [31].

The upstream section of the river is relatively clean, although some agricultural and livestock nonpoint pollution sources exist. In the midstream and downstream sections, the pollution loads rapidly increase due to the presence of large industrial complexes and dense urban areas [32–34]. In addition, in the downstream section of the river, there is concern over the inflow of nonpoint sources during rainfall events from widespread greenhouse farming. Indeed, frequent inflows of high-concentration pollution sources have occurred in the past due to rapid industrialization and urbanization, contributing to a poor water quality status [35,36]. Although water quality has been greatly improved through continuous management, the inflow of various pollutants continues to be a problem. In particular, the industrial complexes within the basin generate large amounts of wastewater treated in large sewage treatment facilities (STFs) and wastewater treatment facilities (WTFs) and subsequently discharged into the river. These effluents introduce a range of pollutants (e.g., non-biodegradable organic matter, nutrients, and heavy metals) into the river as it passes through the urban areas [28,37,38]. Therefore, further water quality management interventions are urgently required to target these pollutants.

*2.2. Water Quality Analysis*

Water quality monitoring was conducted once a month from 2017 to 2020 at six sites (Figure 2) in the Geumho River Basin (Figure 2), data for which were obtained from the Water Environment Information System database (http://water.nier.go.kr: (accessed on: 1 May 2021)). For each of the surface samples collected from each site, 15 water quality variables were measured in accordance with the official test methods for water pollution in Korea [39]. Water temperature (WT), pH, dissolved oxygen (DO), and electrical conductivity (EC) were measured directly (YSI 6600 EDS, Columbus, OH, USA), and BOD, chemical oxygen demand (COD), total organic carbon (TOC), total suspended solids (TSS),

total nitrogen (TN), nitrate-nitrogen (NO$_3$-N), ammonia-nitrogen (NH$_3$-N), TP, phosphate-phosphorus (PO$_4$-P), chlorophyll-*a* (Chl-*a*), and fecal coliform (FC) were determined in the laboratory in accordance with official test methods [39] using samples transported from the field in an icebox. The analysis methods used are listed in Table 1.

**Table 1.** Unit and analysis method of water quality variables.

| Variable | Abbreviation | Unit | Analysis Method |
|---|---|---|---|
| Biochemical oxygen demand | BOD | mg/L | Winkler-azide method (5-day) |
| Chemical oxygen demand | COD | mg/L | Potassium permanganate (KMnO$_4$) |
| Total organic carbon | TOC | mg/L | High-temperature combustion method |
| Total nitrogen | TN | mg/L | Continuous flow analysis (UV/visible spectrometry) |
| Ammonia-nitrogen | NH$_3$-N | mg/L | Continuous flow analysis (UV/visible spectrometry) |
| Nitrate-nitrogen | NO$_3$-N | mg/L | Continuous flow analysis (ion chromatography) |
| Total phosphorus | TP | mg/L | Continuous flow analysis (UV/visible spectrometry) |
| Phosphate-phosphorus | PO$_4$-P | mg/L | Continuous flow analysis (ion chromatography) |
| Chlorophyll-a | Chl-a | mg/m$^3$ | UV/visible spectrometry |
| Total suspended solids | SS | mg/L | Filtration methods (GF-C) |
| Fecal coliform | FC | CFU/100 mL | Membrane filter |

### 2.3. Statistical Analysis

Multivariate analyses of water quality parameters were performed using CA, PCA, FA, and DA. The data for 15 water quality variables collected from 2017 to 2020 from the 6 monitoring sites were used in this study. All data were standardized using z-scores to prevent errors resulting from the different measurement units of each variable. For all statistical calculations and the development of figures, SPSS 24.0, XL-STAT 2020, and Arc-GIS 10.5 were used.

### 2.4. PCA and FA

PCA is a statistical technique used to reduce dimensions by extracting eigenvalues and eigenvectors from a covariance matrix using the correlations among variables [16,40]. An eigenvalue represents the magnitude of variance that can be explained by the principal component; an eigenvalue > 1.0 indicates that one principal component can explain one or more variables. FA generates a new component (varifactor, VF) by rotating the PCA result using the maximum variance method to reduce the influence of the variables of low importance [41,42]. Therefore, eigenvalues $\geq$ 1.0 were extracted and analyzed here. To assess and verify the suitability of the data for FA, Kaiser–Meyer–Olkin (KMO) and Bartlett's tests were first conducted [14,18]. In the FA results, loading values > 0.75 were classified as "strong", and those between 0.5 and 0.75 were classified as "moderate" [43].

### 2.5. CA

CA is a statistical technique used to classify data by analyzing the differences or similarities among observed values. Hierarchical CA can significantly reduce the dimensionality of data by classifying clusters into homogeneous groups. In general, the similarity between samples is measured using the Euclidean distance, and the distance between clusters is measured using Ward's method [15,44,45]. Here, the 15 different characteristics of the study watershed were investigated based on temporal and spatial classification. In addition, the results of the CA were used as the pre-classification values for the DA.

### 2.6. DA and ANOVA

DA is a statistical technique used to derive a group classification discriminant through the explanatory variables of data groups and classify the groups accordingly [10,15]. The technique minimizes classification errors when data are classified into two or more groups by calculating a linear discriminant function and predicts and confirms the group of dependent variables using independent variables derived from quantitative data. In this study, a DA was performed according to the temporal and spatial classifications based on the CA results. The DA was conducted in standard and stepwise modes, and the results were compared and evaluated. ANOVA compares the variability of two or more groups by comparing the variance within the groups [46]. Here, a one-way ANOVA was performed to examine the homogeneity between the stepwise DA groups. A post-hoc analysis was performed using the Scheffe Test to examine the effect of the different temporal and spatial groups on each of the 15 water quality characteristics.

### 2.7. KWQI

The KWQI was calculated and evaluated for each of the temporal and spatial groups of the study watershed, as is utilized by the Ministry of Environment of Korea [47]. The KWQI considers seven variables (pH, DO, EC, WT, TOC, TN, and TP) and was calculated as follows:

$$KWQI = 100 - \sqrt{\frac{F1^2 + F2^2 + F3^2}{3}}$$

where $F1$ is the fraction calculated by dividing the number of water quality variables that violate standards by the number of all measured water quality variables; $F2$ is the fraction calculated by dividing the total number of standard violations by each water quality variable during the measurement period by the total number of measurements; $F3$ is the sum of the factors that fractionalized the degree of each water quality variable for the standards. The KWQI ranges from 0 to 100, with a higher value representing cleaner water. In this assessment, KWQI values were divided into the following five grades: excellent (80–100), good (60–79), fair (40–59), poor (20–39), and very poor (0–19).

## 3. Results and Discussion

### 3.1. PCA and FA

The FA results obtained by conducting the PCA and then rotating the result using the varimax method identified four VFs with an eigenvalue $\geq 1.0$, and the total variance was 72.18% (Table 2). VF1 represented 22.17% of the total variance, and COD and TOC showed a "strong" loading, and BOD, TSS, and Chl-*a* showed a "moderate" loading. VF2 accounted for 20.45% of the total variance, and DO and TP showed a "strong" loading, and WT, PO$_4$-P, and FC showed a "moderate" loading. VF3 represented 18.43% of the total variance, and TN and NH$_3$-N showed a "strong" loading; EC and NO$_3$-N showed a "moderate" loading. VF4 accounted for 11.13% of the total variance, and pH showed a "strong" loading.

The principal factors (PFs) were expressed in a scatter plot using VF1 and VF2 in Figure 3, which are factors that significantly affect the water quality of the river. BOD, Chl-*a*, TSS, TOC, and COD were extracted from PF1, while FC, PO$_4$-P, WT, and TP were extracted from PF2. As shown in Figure 3, the major variables were found to be organic pollutants, substances related to metabolism in the river, phosphorus, and FC. As for organic pollutants, large amounts of non-biodegradable organic matter have been discharged due to recent rapid urban growth, industrialization, and intense human activities [8,48], implying that more stringent regulations for organic pollutants are required. In addition, for urban rivers that pass through metropolitan cities with large populations and high levels of industrialization, the appropriate operation of STFs and WTFs is required given the continuous discharges from certain pollution sources [28,38].

**Table 2.** Variable loads for the major components of all data are based on the principal component analysis (PCA) and factor analysis (FA).

| Variable | Component | | | |
|---|---|---|---|---|
| | VF1 | VF2 | VF3 | VF4 |
| WT | 0.242 | *0.729* | −0.377 | 0.316 |
| pH | 0.062 | −0.019 | −0.018 | **0.787** |
| DO | 0.009 | **−0.784** | 0.188 | −0.288 |
| BOD | *0.606* | 0.068 | 0.324 | 0.474 |
| COD | **0.928** | 0.035 | −0.071 | −0.005 |
| TOC | **0.864** | 0.167 | 0.111 | −0.008 |
| TSS | *0.692* | 0.428 | −0.044 | −0.153 |
| EC | 0.429 | −0.104 | *0.708* | 0.174 |
| TN | −0.066 | 0.201 | **0.873** | 0.157 |
| NH₃-N | 0.007 | 0.047 | **0.854** | −0.032 |
| NO₃-N | 0.070 | −0.312 | *0.604* | −0.186 |
| TP | 0.265 | **0.835** | 0.221 | −0.01 |
| PO₄-P | 0.111 | *0.656* | −0.084 | −0.484 |
| Chl-a | *0.721* | 0.144 | 0.156 | 0.489 |
| FC | 0.116 | *0.636* | 0.156 | −0.235 |
| Eigenvalue | 3.326 | 3.068 | 2.764 | 1.669 |
| Total variance (%) | 22.17 | 20.45 | 18.43 | 11.13 |
| Cumulative variance (%) | 22.17 | 42.62 | 61.05 | 72.18 |

Note(s): KMO test = 0.736 and Bartlett's test = 0.000, indicating that the data were suitable for PCA/FA and that there was a significant relationship between the variables. Bold and italic values represent strong and moderate loadings, respectively. VF, varifactor; WT, water temperature; DO, dissolved oxygen; BOD, biochemical oxygen demand; COD, chemical oxygen demand; TOC, total organic carbon; TSS, total suspended solids; EC, electrical conductivity; TN, total nitrogen; NH3-N, ammonia-nitrogen; NO3-N, nitrate-nitrogen; PO4-P, phosphate-phosphorus; TP, total phosphorus; Chl-a, chlorophyll-a; FC, fecal coliform.

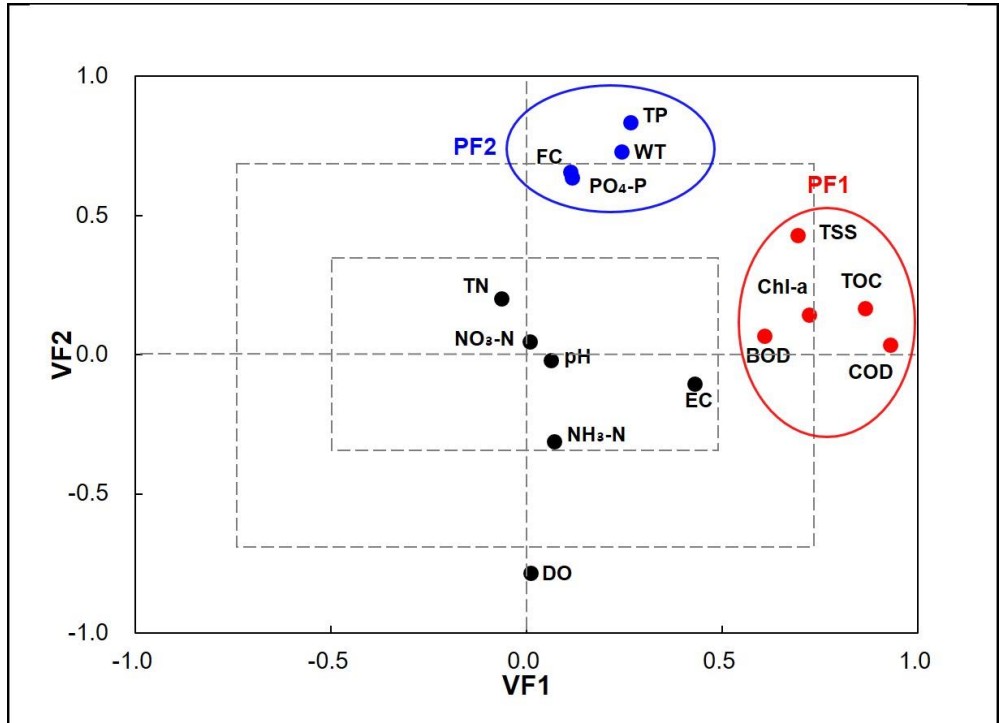

**Figure 3.** Major factors through the scatter plot of VF1 and VF2. Red indicates the principal factor1 and blue indicates the principal factor2. VF, varifactor; PF, principal factor; WT, water temperature; DO, dissolved oxygen; BOD, biochemical oxygen demand; COD, chemical oxygen demand; TOC, total organic carbon; TSS, total suspended solids; EC, electrical conductivity; TN, total nitrogen; NH3-N, ammonia-nitrogen; NO3-N, nitrate-nitrogen; PO4-P, phosphate-phosphorus; TP, total phosphorus; Chl-a, chlorophyll-a; FC, fecal coliform.

*3.2. Water Quality Assessment*

Descriptive statistics (Table 3) for the PF1 and PF2 variables of the PCA/FA results are compared with the environmental standards for rivers in Korea in Table 4. Based on the mean values for PF1, BOD was classified as "very good" and "slightly good" at sites S1 to S3 and "moderate" at sites S4 to S6; COD was "moderate" at sites S1 to S5 and "slightly bad" at site S6; TOC was "slightly good" at sites S1 to S3, "moderate" at sites S4 to S5, and "bad" at site S6. For BOD, COD, and TOC, which represent the levels of organic pollutants, water quality grades varied between each site. This is because biodegradable organic materials are currently more strictly regulated in STFs; however, the regulation of household sewage in urban areas and the non-biodegradable organic matter generated in industrial complexes remains insufficient. In particular, it is assumed that high COD and TOC concentrations at site S6 reflect the large textile-dyeing complexes along the section of the river between sites S5 and S6, from which untreated non-biodegradable organic matter is discharged.

**Table 3.** Range, mean, and standard deviation for variables at the water quality monitoring sites.

| Parameter | | S1 (*n* = 48) | S2 (*n* = 48) | S3 (*n* = 48) | S4 (*n* = 48) | S5 (*n* = 48) | S6 (*n* = 48) |
|---|---|---|---|---|---|---|---|
| BOD (mg/L) | R | 0.3–5.9 | 0.5–5.8 | 1.1–5.0 | 0.9–6.2 | 1.0–5.7 | 1.0–5.7 |
| | M ± SD | 1.0 ± 0.8 | 2.1 ± 1.3 | 2.4 ± 0.9 | 3.0 ± 1.5 | 3.1 ± 1.3 | 3.0 ± 1.3 |
| COD (mg/L) | R | 3.0–7.8 | 3.8–10.3 | 4.3–8.6 | 4.3–9.2 | 4.0–9.8 | 5.8–11.2 |
| | M ± SD | 5.6 ± 1.0 | 6.7 ± 1.4 | 5.7 ± 0.9 | 5.9 ± 1.2 | 5.9 ± 1.4 | 8.2 ± 1.5 |
| TOC (mg/L) | R | 1.7–4.7 | 2.3–5.4 | 2.6–5.9 | 2.5–6.1 | 2.5–5.8 | 4.7–8.9 |
| | M ± SD | 3.4 ± 0.6 | 3.8 ± 0.7 | 3.9 ± 0.7 | 4.1 ± 0.8 | 4 ± 0.8 | 6.3 ± 1.2 |
| TSS (mg/L) | R | 0.6–13.4 | 0.4–23.6 | 1.4–32.8 | 1.8–24.2 | 1.4–26.7 | 2.6–55.0 |
| | M ± SD | 3.3 ± 3.0 | 6.4 ± 5.0 | 4.4 ± 4.5 | 5.0 ± 3.6 | 8.6 ± 5.5 | 10.3 ± 8.3 |
| Chl-*a* (mg/m$^3$) | R | 0.2–18.4 | 1.9–45.9 | 2.8–37.1 | 2.3–78.6 | 2.8–96.7 | 6.7–135.8 |
| | M ± SD | 4.4 ± 3.0 | 10.3 ± 8.2 | 11.5 ± 9.9 | 25.0 ± 20.6 | 27.6 ± 20.9 | 36.3 ± 26.2 |
| TP (mg/L) | R | 0.01–0.08 | 0.01–0.18 | 0.02–0.19 | 0.02–0.16 | 0.02–0.18 | 0.03–0.13 |
| | M ± SD | 0.02 ± 0.01 | 0.04 ± 0.03 | 0.07 ± 0.04 | 0.07 ± 0.04 | 0.08 ± 0.04 | 0.07 ± 0.03 |
| PO$_4$-P (mg/L) | R | 0.00–0.07 | 0.00–0.09 | 0.00–0.11 | 0.00–0.12 | 0.00–0.1 | 0.00–0.1 |
| | M ± SD | 0.01 ± 0.01 | 0.02 ± 0.03 | 0.02 ± 0.03 | 0.01 ± 0.03 | 0.01 ± 0.03 | 0.02 ± 0.03 |
| WT (°C) | R | 3.2–27.6 | 3.8–30.5 | 0.8–30.5 | 2.2–31.9 | 3.6–31.6 | 5.1–29.4 |
| | M ± SD | 14.7 ± 6.8 | 17.0 ± 8.2 | 16.8 ± 8.8 | 17.5 ± 8.9 | 18.1 ± 8.6 | 17.4 ± 7.9 |
| FC (CFU/100 mL) | R | 0–165 | 0–560 | 0–48,000 | 0–20,000 | 0–57,000 | 3–28,963 |
| | M ± SD | 22 ± 40 | 65 ± 133 | 4137 ± 8789 | 2143 ± 4459 | 5559 ± 11,989 | 4224 ± 6497 |

Note(s): S1 to S6, monitoring sites; R, range; M, mean; SD, standard deviation; BOD, biochemical oxygen demand; COD, chemical oxygen demand; TOC, total organic carbon; TSS, total suspended solids; Chl-*a*, chlorophyll-*a*; TP, total phosphorus; PO$_4$-P, phosphate-phosphorus; WT, water temperature; FC, fecal coliform; CFU, colony-forming unit.

TSS was classified as "very good" for all the test sites, with concentrations of ≤25 mg/L. During the flood season, the range and standard deviation of the TSS concentration were higher owing to the large inflows of suspended solids. Although Chl-*a* is not included in the existing water quality standards, Chl-*a* concentrations increased toward the downstream side of the river, where pollution inflows are concentrated relative to the upstream side. Indeed, many of the nutrients introduced from STF effluents stimulate green algae growth in stagnant waters in the downstream section of the river, further deteriorating the river water quality.

Based on PF2, TP was classified as "very good" at sites S1 and S2 and "slightly good" at sites S3 to S6. TP is strictly managed based on the Korean Government's regulatory standards for TP concentrations in STF and WTF effluents [49]. However, continuous monitoring and management is required along the Namcheon tributary, which flows into the river upstream of site S3, as many stables and vineyards are widespread in this area, which can act as nonpoint pollution sources during rainfall events. The PO$_4$-P concentrations were similar to those of TP. The WT varied significantly over time owing to the strong seasonal effect of the monsoon climate. FC concentrations were classified as "good" at sites S1 to S2 but were very high at sites S3 to S6, exceeding the current

environmental standard (1000 colony-forming units (CFU)/100 mL). The increase in FC concentrations on the downstream side likely reflects the introduction of STF and WTF effluents from the metropolitan areas downstream of site S2.

**Table 4.** Environmental standards for river water quality in Korea [50].

| Grade | BOD (mg/L) | COD (mg/L) | TOC (mg/L) | TSS (mg/L) | TP (mg/L) | FC (CFU/100 mL) |
|---|---|---|---|---|---|---|
| Very good | 1.0 or less | 2.0 or less | 2.0 or less | 25 or less | 0.02 or less | 10 or less |
| Good | 1.0–2.0 | 2.0–4.0 | 2.0–3.0 | 25 or less | 0.02–0.04 | 10–100 |
| Slightly good | 2.0–3.0 | 4.0–5.0 | 3.0–4.0 | 25 or less | 0.04–0.1 | 100–200 |
| Moderate | 3.0–5.0 | 5.0–7.0 | 4.0–5.0 | 25 or less | 0.1–0.2 | 200–1000 |
| Slightly bad | 5.0–8.0 | 7.0–9.0 | 5.0–6.0 | 100 or more | 0.2–0.3 | - |
| Bad | 8.0–10.0 | 9.0–11.0 | 6.0–8.0 | - | 0.3–0.5 | - |
| Very bad | Over 10.0 | Over 11.0 | Over 8.0 | - | Over 0.5 | - |

Note(s): BOD, biochemical oxygen demand; COD, chemical oxygen demand; TOC, total organic carbon; TSS, total suspended solids; TP, total phosphorus; FC, fecal coliform; CFU, colony-forming unit.

When water quality was assessed by comparing the major variables extracted from the PCA and FA with the current environmental standards, overall water quality was poor in the downstream section of the river relative to the upstream section because of the significant influence of the point pollution sources located in the urban areas, most notably the STFs and WTFs. In addition, the water quality characteristics of the river were found to vary depending on the surrounding land-use patterns.

### 3.3. Temporal and Spatial Cluster Analysis

The results of the CA and temporal and spatial classification using the water quality data of the monitoring sites (S1 to S6) are shown in Figure 4. Based on the seasonal CA results (Figure 4a), March to June was classified as "cluster 1 (spring)", July to September as "cluster 2 (summer)", October to December as "cluster 3 (autumn)", and January to February as "cluster 4 (winter)." These four distinct seasons reflect the mid-latitude temperate monsoonal climate of the study basin. The average total precipitation of the target watershed over the last 10 years was 1089.1 mm, 54.6% (594.8 mm) of which fell between July and September. Due to this strong temporal rainfall bias, the flow of the river significantly varies between seasons, which is also assumed to result in seasonal water quality variations.

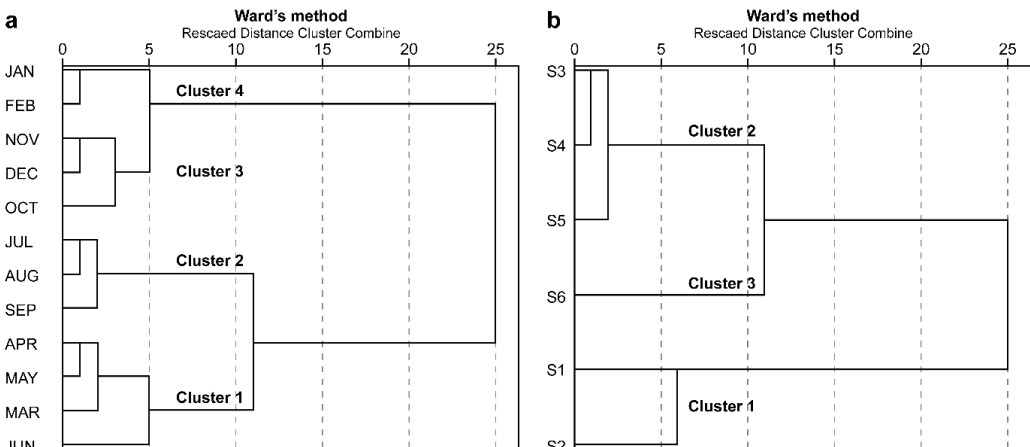

**Figure 4.** Dendrograms that show seasonal and spatial clusters. (**a**): seasonal cluster analysis; (**b**): spatial cluster analysis.

Based on the spatial CA results (Figure 4b), sites S1 to S2 were classified as "cluster 1 (upstream: US)", sites S3 to S5 as "cluster 2 (midstream: MS)", and site S6 as "cluster 3 (downstream: DS)." These clusters correspond to the US section with relatively few pollution sources; the MS section with urban STFs; and the DS section with large industrial complexes, WTFs, and greenhouse facilities. These areas are more clearly distinguished from the land-use map of the Geumho River watershed (Figure 5). Cluster 1 corresponds to the area with a high proportion of forests and farmlands; cluster 2 corresponds to urban areas with residential, commercial, and industrial land-cover types; and cluster 3 includes a complex mix of land uses (Figure 5).

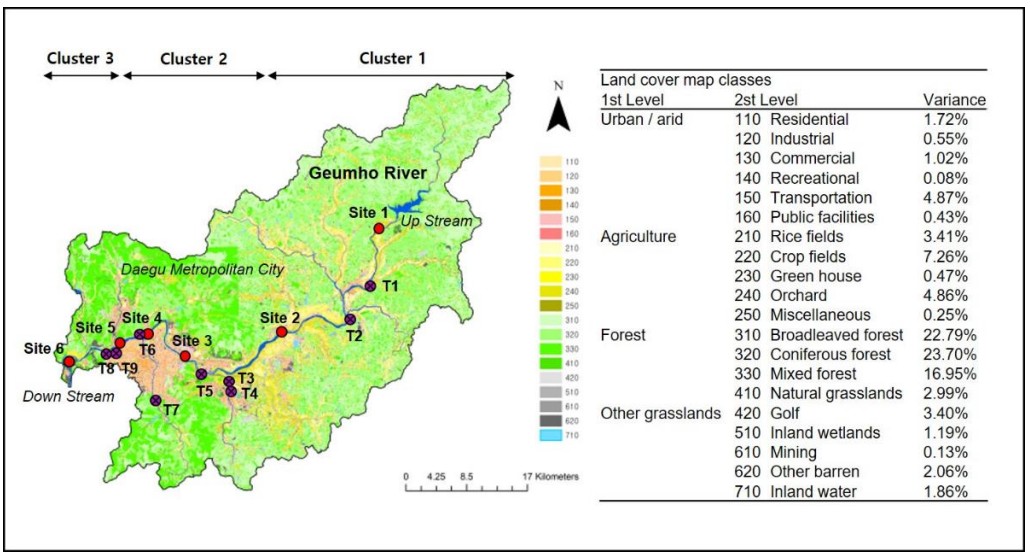

**Figure 5.** Land cover map of the target watershed drawn using Arc-GIS. Land cover codes 110 to 160 are urban (orange series), 210 to 250 are agriculture (yellow series), 310 to 330 are forest (green series), etc. Red point, monitoring sites (S1 to S6); purple point, sewage and wastewater treatment facilities (T1 to T9).

### 3.4. Temporal and Spatial DA

A DA was conducted following the CA according to the pre-classified temporal and spatial water quality characteristics. Thus, temporal accuracy was maximized by conducting the DA according to the seasonal classification (spring, summer, autumn, and winter). The DA was conducted in standard and stepwise modes, and the discriminant function and classification matrix constructed for each mode are shown in Tables 5 and 6. In the standard mode, all 15 water quality variables were used, and the average accuracy of the classified matrix was as high as 80.2% (Table 6). In the stepwise mode, which reduces the number of variables relative to the standard mode, six water quality variables (WT, BOD, TOC, EC, $NH_3$-N, and TP) were used, yielding an accuracy of 78.8% (Table 6). Notably, in the stepwise mode, spring and autumn showed relatively low classification accuracies of 75.0% and 74.0%, respectively. To evaluate water quality according to the temporal classification, the six water quality variables identified based on the stepwise approach are shown in a box plot in Figure 6, which confirmed the homogeneity of the groups based on the ANOVA results and the Scheffe Test ($p < 0.05$).

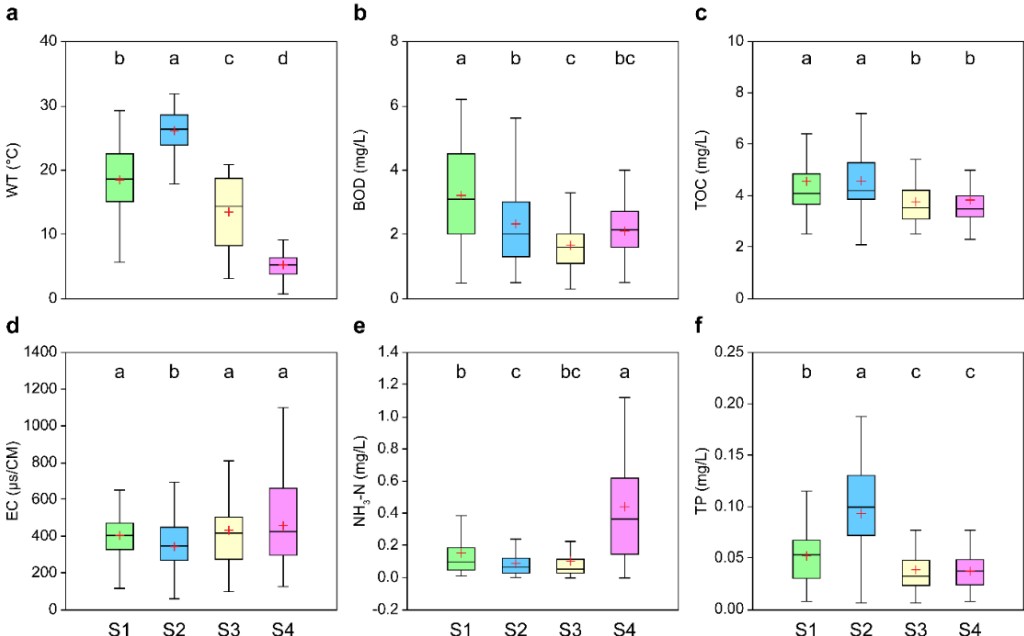

**Figure 6.** Box plots of the temporal stepwise mode discriminant function. The middle line of each box represents the median value, the lower line represents the first quartile (25%), the upper line represents the third quartile (75%), the lower bar represents the minimum (median-1.5 × interquartile range; IQR), and the upper bar represents the maximum (median + 1.5 × IQR); the red cross mark indicates the average value. IQR is the third quartile minus the first quartile, and the letters above the box plot represent the results of the Scheffe Test ($p < 0.05$). (**a**): water temperature; (**b**): biochemical oxygen demand; (**c**): total organic carbon; (**d**): electrical conductivity; (**e**): ammonia-nitrogen; (**f**): total phosphorus. S1, spring; S2, summer; S3, autumn; S4, winter.

**Table 5.** Discriminant functions to determine temporal changes (Fisher's linear discriminant functions).

| Variable | Standard Mode | | | | Stepwise Mode | | | |
|---|---|---|---|---|---|---|---|---|
| | Spring | Summer | Autumn | Winter | Spring | Summer | Autumn | Winter |
| WT | 1.403 | 1.835 | 1.245 | 0.832 | 0.839 | 1.153 | 0.618 | 0.24 |
| pH | 30.538 | 29.493 | 30.018 | 30.593 | | | | |
| DO | 4.33 | 4.58 | 4.534 | 4.707 | | | | |
| BOD | −0.539 | −1.394 | −1.974 | −2.142 | 0.918 | −0.156 | −0.334 | −0.257 |
| COD | 4.336 | 4.012 | 3.849 | 4.315 | | | | |
| TOC | 3.706 | 3.533 | 2.887 | 3.138 | 2.63 | 2.639 | 1.93 | 2.402 |
| TSS | −0.28 | −0.306 | −0.199 | −0.309 | | | | |
| EC | −0.023 | −0.024 | −0.015 | −0.018 | −0.004 | −0.005 | 0.004 | 0.001 |
| TN | 0.977 | 0.838 | 0.956 | 0.978 | | | | |
| NH$_3$-N | 2.342 | 2.842 | 2.574 | 2.226 | 5.91 | 6.678 | 3.798 | 12.755 |
| NO$_3$-N | 11.22 | 11.992 | 9.251 | 18.392 | | | | |
| TP | 10.849 | 30.924 | 7.444 | 37.27 | −26.276 | 17.998 | −15.883 | −8.094 |
| PO$_4$-P | 23.452 | 62.642 | 41.274 | 38.946 | | | | |
| Chl-a | −0.467 | −0.436 | −0.425 | −0.415 | | | | |
| FC | 0.000 | 0.000 | 0.000 | 0.000 | | | | |
| (Constant) | −183.986 | −185.44 | −174.56 | −181.562 | −15.585 | −22.645 | −9.746 | −9.174 |

Note(s): Spring, March–June; summer, July–September; autumn, October–December; winter, January–February; WT, water temperature; DO, dissolved oxygen; BOD, biochemical oxygen demand; COD, chemical oxygen demand; TOC, total organic carbon; TSS, total suspended solids; EC, electrical conductivity; TN, total nitrogen; NH$_3$-N, ammonia-nitrogen; NO$_3$-N, nitrate-nitrogen; PO$_4$-P, phosphate-phosphorus; TP, total phosphorus; Chl-*a*, chlorophyll-*a*; FC, fecal coliform.

**Table 6.** Classification matrix for the discriminant analysis of temporal changes.

| | Monitoring Clusters | Correct (%) | Regions Assigned by DA | | | | |
|---|---|---|---|---|---|---|---|
| | | | Spring | Summer | Autumn | Winter | Total |
| Standard mode | Spring | 76.0 | 73 | 6 | 12 | 5 | 96 |
| | Summer | 87.6 | 7 | 63 | 2 | 0 | 72 |
| | Autumn | 72.2 | 9 | 4 | 52 | 7 | 72 |
| | Winter | 89.6 | 1 | 0 | 4 | 43 | 48 |
| | Total | 80.2 | | | | | |
| Stepwise mode | Spring | 74.0 | 71 | 9 | 13 | 3 | 96 |
| | Summer | 81.9 | 12 | 59 | 1 | 0 | 72 |
| | Autumn | 75.0 | 9 | 4 | 54 | 5 | 72 |
| | Winter | 89.6 | 0 | 0 | 5 | 43 | 48 |
| | Total | 78.8 | | | | | |

Note(s): Spring, March to June; summer, July to September; autumn, October to December; winter, January to February; DA, discriminant analysis

WT was the highest in summer and tended to fluctuate in line with seasonal changes. WT variability was higher in spring and autumn than in summer and winter. BOD was the highest and more variable in spring and tended to reduce in summer and autumn, increasing again during winter. In contrast, TOC concentrations were the highest in summer compared to the other seasons. BOD and TOC are representative organic pollution indicators, and the fluctuations in their concentrations likely reflect the effect of increased river flows due to rainfall. Although average EC values did not significantly differ across the seasons, there were seasonal differences in EC variability. $NH_3$-N concentrations were significantly higher in the winter season than during the other seasons (Figure 6). This suggests high $NH_3$-N discharges from STFs and WTFs coupled with relatively low denitrification rates during winter, when water temperatures are lower. In many countries, $NH_3$-N is managed by setting water quality standards for rivers and the effluents from water treatment facilities [51]; however, $NH_3$-N is not regulated in Korea. $NH_3$-N regulation requires further consideration in the context of water quality management in Korea. For example, aquatic ammonia toxicity in rivers increases when the $NH_3$-N concentrations rise above 1.0 mg/L and pH > 8.0 [52]; in winter, the concentration of $NH_3$-N in the Geumho River was >1.0 mg/L (Figure 6), and the average pH in the MS section was 8.5 (Figure 7). These results suggest that $NH_3$-N concentrations may impact aquatic life in this river [53]. In addition, inflows containing high concentrations of $NH_3$-N have negative impacts on BOD and eutrophication in rivers [51,54]. In the case of TP, average concentrations and ranges were much higher in summer than during the other seasons. This likely reflects the various nonpoint sources that contribute pollutants during the summer flood season as well as untreated effluents that exceed the capacity of the STFs during intense rainfall (i.e., storm overflow).

A DA was also conducted based on the CA spatial classification (US, MS, and DS). The discriminant function and classification matrix of the results following the standard and stepwise modes are shown in Tables 7 and 8, respectively. In the standard mode, all 15 variables were used, and the average accuracy of the classified matrix was as high as 97.9% (Table 8). In the stepwise mode, seven variables (pH, BOD, COD, TOC, EC, TN, and TP) were used, yielding a classification accuracy of 92.0% (Table 8). Notably, the classification accuracy for US (89.6%) was relatively low compared to other classification results in the stepwise mode, and the accuracies for MS and DS were 91.7% and 97.9%, respectively.

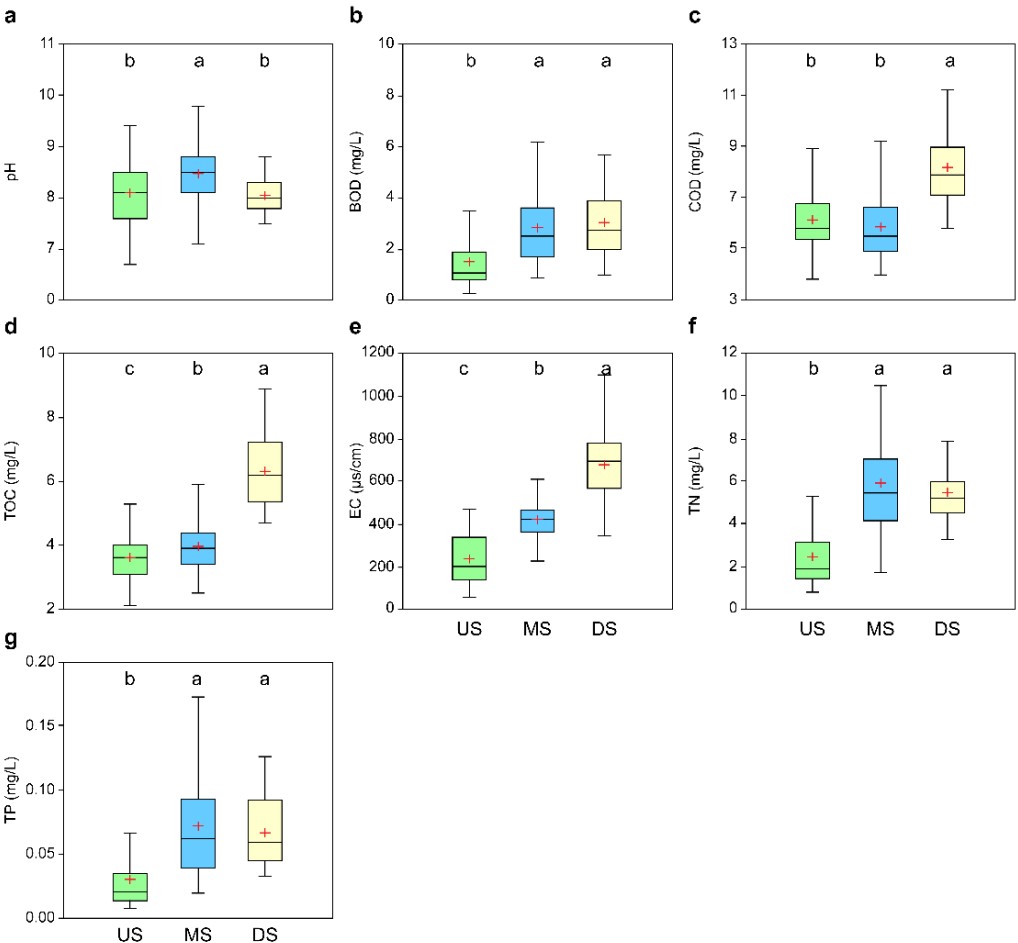

**Figure 7.** Box plots of the spatial stepwise mode discriminant function. The middle line of each box represents the median value, the lower line represents the first quartile (25%), the upper line represents the third quartile (75%), the lower bar represents the minimum (median-1.5 × interquartile range; IQR), and the upper bar represents the maximum (median + 1.5 × IQR); the red cross mark indicates the average value. The IQR is the third quartile minus the first quartile, and the letters above the box plot represent the results of the Scheffe Test ($p < 0.05$). (**a**): pH; (**b**): biochemical oxygen demand; (**c**): chemical oxygen demand; (**d**): total organic carbon; (**e**): electrical conductivity; (**f**): total nitrogen; (**g**): total phosphorus. US, upstream; MS, midstream; DS, downstream.

To evaluate the water quality characteristics according to the spatial classification, the seven water quality variables identified based on the stepwise analysis are shown in Figure 7, which confirms the homogeneity of the groups based on the ANOVA results and the Scheffe Test ($p < 0.05$). pH, TN, and TP were high in the MS, which likely reflects nutrient inputs from the six STFs and WTFs located between sites S3 and S6 (Figure 2). These inputs can stimulate phytoplankton growth, which can decrease oxygen saturation and increase carbon dioxide saturation in water, thereby increasing pH [55]. BOD, COD, TOC, and EC increased toward the DS section of the river compared to the US section, suggesting the inflow of organic pollutants in association with urbanization and industrialization in the DS area (see Figure 5). TN and TP also showed similar trends, although average concentrations in the MS section were similar to the DS section, and variability was higher in the MS section than in the DS section. These patterns may reflect the high proportion of farmland in the MS section relative to the DS section, which acts as a nonpoint pollution source during certain conditions. To create an urban landscape, the amount of water through river dredging increased. However, the flow of the river decreased, and stagnant water formed. This reduction in the self-purification capacity of the river has an adverse effect on water quality. In addition, as a result of reductions in the flow during the dry season,

effluents from the STFs and WTFs typically make up the majority of instream flow [32]. The treatment facilities in the Geumho River Basin treat both domestic wastewater and industrial effluents, which can contain non-biodegradable organic matter, and require ongoing regulation and management [48].

**Table 7.** Discriminant functions to determine spatial changes (Fisher's linear discriminant functions).

| Variable | Standard Mode | | | Stepwise Mode | | |
|---|---|---|---|---|---|---|
| | **US** | **MS** | **DS** | **US** | **MS** | **DS** |
| WT | 1.174 | 0.868 | 0.908 | | | |
| pH | 31.435 | 34.117 | 29.45 | 30.331 | 32.07 | 28.173 |
| DO | 4.516 | 3.268 | 3.639 | | | |
| BOD | −2.012 | −1.379 | −2.893 | −4.808 | −4.121 | −5.55 |
| COD | 3.587 | 1.51 | 1.918 | 2.926 | 0.933 | 1.163 |
| TOC | 1.177 | 2.96 | 7.835 | 0.065 | 1.552 | 6.727 |
| TSS | −0.168 | −0.16 | −0.231 | | | |
| EC | −0.021 | −0.011 | 0.016 | −0.004 | 0.009 | 0.032 |
| TN | 1.042 | 1.394 | 1.097 | 2.234 | 2.802 | 2.575 |
| $NH_3$-N | 2.467 | 3.275 | 3.911 | | | |
| $NO_3$-N | 11.52 | 14.103 | 8.392 | | | |
| TP | 38.524 | 103.418 | 17.876 | 69.724 | 131.491 | 98.784 |
| $PO_4$-P | 25.941 | 17.324 | 92.628 | | | |
| Chl-*a* | −0.438 | −0.396 | −0.305 | | | |
| FC | 0.000 | 0.000 | 0.000 | | | |
| (Constant) | −176.379 | −186.667 | −189.161 | −132.62 | −151.867 | −153.374 |

Note(s): US, upstream (S1 to S2 site); MS, midstream (S3 to S5 site); DS, downstream (S6 site); WT, water temperature; DO, dissolved oxygen; BOD, biochemical oxygen demand; COD, chemical oxygen demand; TOC, total organic carbon; TSS, total suspended solids; EC, electrical conductivity; TN, total nitrogen; $NH_3$-N, ammonia-nitrogen; $NO_3$-N, nitrate-nitrogen; $PO_4$-P, phosphate-phosphorus; TP, total phosphorus; Chl-*a*, chlorophyll-*a*; FC, fecal coliform.

**Table 8.** Classification matrix for the discriminant analysis of spatial changes.

| Period | Monitoring Clusters | Correct (%) | Regions Assigned by DA | | | |
|---|---|---|---|---|---|---|
| | | | **US** | **MS** | **DS** | **Total** |
| Standard mode | US | 96.9 | 93 | 3 | 0 | 96 |
| | MS | 97.9 | 3 | 141 | 0 | 144 |
| | DS | 100 | 0 | 0 | 48 | 48 |
| | Total | 97.9 | | | | |
| Stepwise mode | US | 89.6 | 86 | 10 | 0 | 96 |
| | MS | 91.7 | 8 | 132 | 4 | 144 |
| | DS | 97.9 | 0 | 1 | 47 | 48 |
| | Total | 92.0 | | | | |

Note(s): US, upstream (S1 to S2 site); MS, midstream (S3 to S5 site); DS, downstream (S6 site); DA, discriminant analysis.

### 3.5. Water Quality Assessment Using the KWQI

The water quality of the river was assessed for each temporally and spatially classified group using seven water quality variables applied by the Korean Ministry of Environment (pH, DO, EC, WT, TOC, TN, and TP) (Table 9 and Figure 8). For the seasonal groups, KWQI values ranged from 55.2 to 61.6 in spring, from 58.4 to 70.1 in summer, from 60.4 to 66.3 in autumn, and from 67.7 to 75.1 in winter. In addition, the average KWQI values for these seasons across the four-year study period were 59.2, 63.4, 63.1, and 70.9, respectively. The average KWQI value was lowest in spring (59.2), falling below 60 during the 2017–2018 period.

**Table 9.** Seasonal and spatial WQI scores during the research period.

| Year | Seasonal | | | | Spatial | | |
|------|--------|--------|--------|--------|------|------|------|
| | Spring | Summer | Autumn | Winter | US | MS | DS |
| 2017 | **55.2** | 60.8 | 60.4 | 75.1 | 64.4 | **58.9** | 60.3 |
| 2018 | **59.9** | 64.2 | 66.3 | 67.7 | 68.3 | **53.5** | 61.6 |
| 2019 | 60.1 | **58.4** | 60.4 | 70.3 | 68.5 | **53.3** | 61.4 |
| 2020 | 61.6 | 70.1 | 65.6 | 70.6 | 69.0 | **59.8** | 66.6 |
| Mean | **59.2** | 63.4 | 63.1 | 70.9 | 67.5 | **56.4** | 62.5 |

Note(s): WQI scores at the "Fair" grade (<60) are shown in bold. Spring, March to June; summer, July to September; autumn, October to December; winter, January to February; US, upstream (S1 to S2 site); MS, midstream (S3 to S5 site); DS, downstream (S6 site).

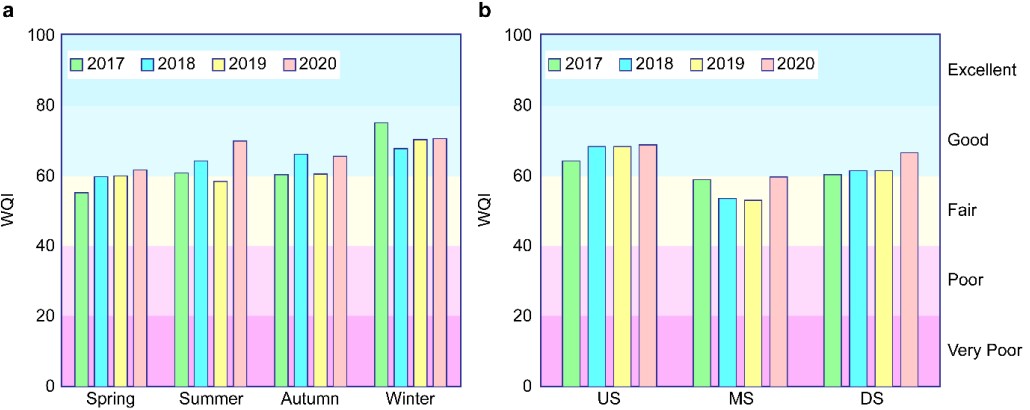

**Figure 8.** Seasonal and spatial WQI scores. Four bar graphs in each seasonal or spatial group show WQI scores from 2017 to 2020 from the left to right. The letters on the right represent the water quality grades of the WQI. Excellent, Suitable for hydrophilic activities with clean water; Good, Suitable for hydrophilic activities with good water; Fair, sometimes pollutants may be introduced and affect hydrophilic activity; Poor, Attention of hydrophilic activities due to frequent inflow of pollutants; Very poor, Inadequate for hydrophilic activities with a high water pollution level. WQI, water quality index; (**a**), seasonal WQI; (**b**), spatial WQI.

For spatial groups, KWQI values ranged from 64.4 to 69.0 in the US section, from 53.3 to 59.8 in the MS section, and from 60.3 to 66.6 in the DS section. In addition, the average KWQI values for these sections across the four-year study period were 67.5, 56.4, and 62.5, respectively. The Korean Ministry of Environment considers that there is a pollutant inflow problem when the KWQI is ≤60 [43]. The MS group showed the lowest average KWQI value (56.4), indicating that intensive water quality management measures are required in this section of the river. The KWQI evaluation also showed that pollution was greatest during seasons when instream flows were low and, in those sections, where multiple treatment facilities are located, which is consistent with the results discussed in Section 3.4.

## 4. Conclusions

Major water quality variables for the Geumho River were temporally and spatially classified and analyzed using various statistical analysis techniques.

Among the variables extracted from the PCA and FA, COD, TOC, TP, and TSS exhibited very strong loadings, indicating that non-biodegradable organic matter and nonpoint sources generated during rainfall events are the main causes of water quality deterioration in the study basin. River water quality dynamics during the study period were evaluated based on the temporal (spring, summer, autumn, and winter) and spatial (US, MS, and DS) classifications, and countermeasures were identified. According to the temporal and spatial discriminant analyses, temporally important variables were WT, BOD, TOC, EC, $NH_3$-N, and TP, and spatially important variables were pH, BOD, COD, TOC, EC, TN, and TP. These variables were higher in spring and summer than in other seasons. On a special scale, the

water quality parameters that are indicators of the pollutant load in river water increased from the US to the DS side. Based on KWQI, the overall water quality classification for the study river is "good", although this drops to "fair" during some seasons and in some sections. The poorest water quality occurs in spring (average KWQI = 59.2) and in the MS section (average KWQI = 56.4).

Given the critical importance of rivers for human life, including their roles in leisure activities and industrial activities, efficient water quality management is required to minimize adverse impacts on human and environmental health. As this study does not include the results of the investigation of treatment facilities located in downtown areas, it is judged that it will be helpful for water quality management and countermeasures planning if the additional data on treated effluent discharge concentrations for treatment facilities and rainfall data are added and analyzed in future studies. Based on the results obtained in this study, future research and monitoring should target those seasons and river sections identified as having the lowest water quality, i.e., the spring season and the MS and DS river sections.

**Author Contributions:** All authors contributed to the planning and structuring of the study. C.J.: manuscript writing, data analysis, and visualization. H.K.: manuscript review and editing. S.K.: data collection and investigation. All authors have read and agreed to the published version of the manuscript.

**Funding:** This work was supported by the National Institute of Environmental Research (NIER), funded by the Ministry of Environment (MOE) of the Republic of Korea [grant number NIER-2022-01-01-043]. The funding agency had no role in study design; in the collection, analysis, and interpretation of data; in the writing of the report; and in the decision to submit the article for publication.

**Institutional Review Board Statement:** Not applicable.

**Informed Consent Statement:** Not applicable.

**Data Availability Statement:** The datasets used and/or analyzed during the current study are available from the corresponding author upon reasonable request.

**Acknowledgments:** We thank the anonymous reviewers for their comments on improving the manuscript. We also thank the members of the Nakdong River Environmental Research Center for their assistance.

**Conflicts of Interest:** The authors declare that they have no conflict of interest.

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
