# Peer review of "Temporal and Spatial Water Quality Assessment of the Geumho River, Korea, Using Multivariate Statistics and Water Quality Indices"

_water, doi:10.3390/w14111761_

Round 1

Reviewer 1 Report

As per the suggestions author improve the whole paper, I have suggested the major revision. I have enclosed my details comments in attachment.

Author Response

We wrote a response to the reviewer opinion in the attached file.

Reviewer 2 Report

In the manuscript, the authors applied various techniques of multivariate analysis on a data set concerning monthly measurements (from 2017 to 2020) of 15 variables at 6 monitored sites. Using these techniques, the independence among cases is assumed; therefore, I suggest to clarify this hypothesis and to highlight the adequacy (suitability) of this hypothesis for the data set under study.

Author Response

(The authors gave the same response as above.)

Round 2

Reviewer 1 Report

Revised versions paper have included suggestios by authors.. Now current version are done not required any changes, I have accepted the article for publication.